# Adversarial Style Augmentation for Domain Generalized Urban-Scene Segmentation

## Abstract

In this paper, we consider the problem of domain generalization in semantic segmentation, which aims to learn a robust model using only labeled synthetic (source) data. The model is expected to perform well on unseen real (target) domains. Our study finds that the image style variation can largely influence the model's performance and the style features can be well represented by the channel-wise mean and standard deviation of images. Inspired by this, we propose a novel adversarial style augmentation (**AdvStyle**) approach, which can dynamically generate hard stylized images during training and thus can effectively prevent the model from overfitting on the source domain. Specifically, AdvStyle regards the style feature as a learnable parameter and updates it by adversarial training. The learned adversarial style feature is used to construct an adversarial image for robust model training. AdvStyle is easy to implement and can be readily applied to different models. Experiments on two synthetic-to-real semantic segmentation benchmarks demonstrate that AdvStyle can significantly improve the model performance on unseen real domains and show that we can achieve the state of the art. Moreover, AdvStyle can be employed to domain generalized image classification and produces a clear improvement on the considered datasets.

## 1 Introduction

Semantic segmentation plays a critical role in autonomous driving, which has achieved impressive improvements with the recent development of deep segmentation networks (Long et al., 2015; Chen et al., 2018a; Badrinarayanan et al., 2017). However, these achievements have been mostly attributed to large-scale labeled segmentation datasets, in which annotating pixel-wise labels is very expensive and time-consuming. In addition, the model trained on one dataset commonly produces poor performance on unseen datasets captured in different conditions. This degradation phenomenon is mainly caused by domain shifts (Choi et al., 2021), including differences in weather, season, light, category statics, etc. For instance, the segmentation model trained on the dataset captured in sunny London will have low accuracy when deployed on the streets of Zurich in rainy weather.

To address the cross-domain problem, domain adaptation methods (Tsai et al., 2018; Vu et al., 2019; Luo et al., 2019; Zhang et al., 2021) are designed to transfer the knowledge of labeled source data to unlabeled target data. However, one of their main drawbacks is that they require the use of target data during training, which cannot always be accessible in practice. Another promising line is domain generalization (DG), which focuses on learning a generalizable model using only the labeled source domain. To reduce the annotating cost and protect data privacy, the existing DG works (Choi et al., 2021; Yue et al., 2019) in semantic segmentation commonly choose to learn the robust model with synthetic data, *e.g.*, GTAV (Richter et al., 2016). In this paper, we focus on this synthetic-to-real DG problem for semantic segmentation.

To eliminate the impact caused by the large domain gap between synthetic and real data, existing solutions mainly aim at augmenting the synthetic source data with extra real-world samples (Yue et al., 2019; Huang et al., 2021) or learning domain-invariant features with carefully designed modules (Pan et al., 2018; Choi et al., 2021). The key idea behind them is to avoid the model overfitting on the source domain. This work follows this idea and introduces a new augmenting approach in the perspective of image style for domain generalized semantic segmentation, which is motivated by the observations in Fig. 1. *First*, when showing the samples of different datasets in Fig. 1(a) we observe

CityScapes BDD Mapillary GTAV Random Noise (↓16% mIoU) C-Style (↓20% mIoU)

Change Style

B-Style (↓17% mIoU) M-Style (↓13% mIoU)

(a) Examples of different datasets. (b) Examples of changing styles for GTAV.

Figure 1: (a) Examples of different datasets. The image styles from different datasets are commonly very different. (b) Examples of changing style feature for a GTAV sample, including adding random noise and replacing the style feature with one of samples from the other datasets. The mIoU performance is largely reduced when applying the four style variations to the GTAV testing set.

that the image styles are quite different among them, *e.g.*, the road color. *Second*, the channel-wise mean and standard deviation of an image, which is called style feature in this paper, can well represent the image style. When changing the style feature, the image style of an example varies while the semantic content is well maintained (see Fig. 1(b)). *Third*, changing the style features of testing samples will largely deteriorate the model performance (see the numbers in Fig. 1(b)). This indicates that the model performance is highly related to the style distributions of the testing set.

Taking the above observations into consideration, we argue that the image style is an important factor that affects the model performance and propose the adversarial style augmentation (**AdvStyle**) for domain generalized semantic segmentation. Specifically, AdvStyle contains two steps: adversarial style learning and robust model learning. In adversarial style learning, we first decompose the training sample into style feature and normalized image. The style feature is regarded as a learnable parameter, which is used to reconstruct a new training example together with the normalized image. Then, we feed the reconstructed example into the segmentation model and optimize the style feature using the adversarial segmentation loss. The updated style feature is called adversarial style feature and is used to produce hard example in the following step. In robust model training, we first generate an adversarial example by de-normalizing the normalized image with the learned adversarial style feature. The adversarial image and the original image are then used to train a robust model using the segmentation loss. In AdvStyle, the adversarial image is dynamically generated based on the current model. In this way, the model is always encouraged to update with difficult styles and thus will be more robust to style variations in unseen domains. In Fig. 2, we show the comparison between AdvStyle and traditional augmentation methods. Our AdvStyle can significantly improve the model performance on unseen target domains and clearly outperforms the other augmentation methods. To summarize, our contributions are threefold:

- We propose the novel adversarial style augmentation (AdvStyle) for domain generalized semantic segmentation, which can consistently improve the results on unseen real domains. AdvStyle introduces very limited learnable parameters (6-dim feature for each example) and can be easily implemented with different networks and DG methods.
- Experiments on two synthetic-to-real DG benchmarks demonstrate the effectiveness of the proposed AdvStyle and show that we achieve new state-of-the-art DG performance.
- We show that AdvStyle can also be applied to single DG in image classification and can produce state-of-the-art accuracy on two datasets.

Original (mIoU=21%) Gaussian Blur (mIoU=25%) ◀——————— **AdvStyle (mIoU=37%)** ———————▶

Color Jittering (mIoU=26%) Blur+Jittering (mIoU=28%) ◀——————— **AdvStyle+Blur+Jitter (mIoU=39%)** ———————▶

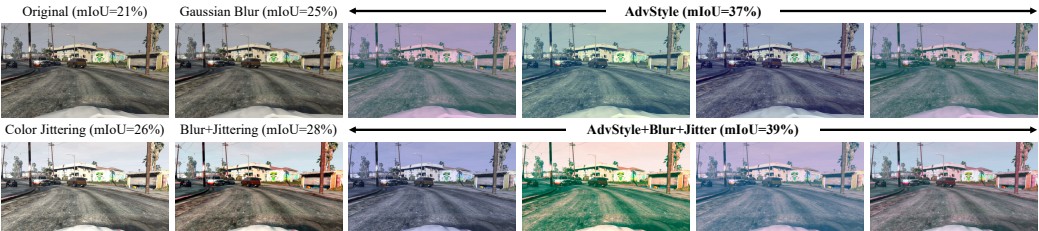

Figure 2: Illustration of different data augmentation methods. We use GTA5 as the source domain and the ResNet-50 as the backbone. The mIoU given in parentheses is evaluated on CityScapes validation set for the model trained with the corresponding augmentation method.

## 2 RELATED WORK

**Domain Generalization (DG) in Semantic Segmentation.** To tackle the deficiency of annotated segmentation data, DG is introduced to learn a robust model with one or multiple source domains, where the model is expected to perform well on unseen domains. Recent works mostly use synthetic data as the source domain, which can be automatically generated but have a large distribution gap to real-world datasets. One main stream of DG methods (Yue et al., 2019; Huang et al., 2021) focuses on augmenting training samples with extra real-world data from ImageNet (Deng et al., 2009). Learning domain-invariant features (Choi et al., 2021; Pan et al., 2018; Tang et al., 2021) is another stream to narrow the domain gap. Pan et al. (2018) and Choi et al. (2021) leverage instance normalization (Ulyanov et al., 2016) and whitening transformation to remove the domain-specific information, respectively. Tang et al. (2021) exchanges style features of two samples and adjusts style features with the attention mechanism. Different from the above methods, our AdvStyle generates new samples by learning adversarial styles using only the synthetic source data.

**Adversarial Training in DG.** Adversarial training (Goodfellow et al., 2015) is initially proposed to learn a robust model that can combat imperceptible perturbations. In recent years, adversarial training is applied to single DG in image classification (Volpi et al., 2018; Qiao et al., 2020; Qiao & Peng, 2021; Fan et al., 2021), by regarding adversarial samples as augmented unseen samples. Volpi et al. (2018) is the first to introduce adversarial samples in DG by max-min iterative training procedure. Later methods form novel domains with the generated adversarial samples and learn a domain-invariant representation by meta-learning (Qiao et al., 2020; Qiao & Peng, 2021) or adaptive normalization (Fan et al., 2021). Different from them, this paper adopts adversarial training for semantic segmentation and generates adversarial samples in the perspective of image style.

**Style Variation.** Style features are widely studied in image translation (Huang & Belongie, 2017; Dumoulin et al., 2017). By varying the style features, the image style can be changed while semantic content will be maintained. Inspired by this, recent works focus on generating data of novel distributions by modifying style features, which are used to train a more robust model. One effective manner is to generate new styles by exchanging (Zhou et al., 2021; Zhao et al., 2021) or mixing styles (Tang et al., 2021) between samples. On the other hand, new styles can be generated by learnable modules (Wang et al., 2021). Instead, we generate novel styles by adversarial training, which encourages the model to always optimize with hard stylized examples. This work is also closely related to Bhattad et al. (2020), which generates adversarial examples by colorizing. However, it requires a pre-trained colorization model to change the image color, which is much more complex than our AdvStyle. In addition, Bhattad et al. (2020) aims to impair the performance of models by adversarial examples. In contrast, our AdvStyle leverages the adversarial examples to improve the generalization ability of the segmentation model. This work also has a connection with "Learning-to-Simulate" (Ruiz et al., 2019). However, Ruiz et al. (2019) tries to learn good sets of parameters for an image rendering simulator in actual computer vision applications while we attempt to learn a generalized model for semantic segmentation.

## 3 METHOD

**Problem Definition.** Synthetic-to-real domain generalization (DG) focuses on training a robust model with *one* labeled synthetic domain $\mathcal{S}$, where the model is expected to perform well on unseen domains $\{\mathcal{T}_1, \mathcal{T}_2, \cdots\}$ of different real-world distributions. As stated by Volpi et al. (2018), the DG task can be formulated as solving the worst-case problem:

$$\min_{\theta} \sup_{\mathcal{T}:D(\mathcal{S},\mathcal{T})\leq\rho} \mathbb{E}_T \left[ \mathcal{L}_{\text{task}}\left(\theta; \mathcal{T}\right) \right], \tag{1}$$

where $\theta$ is the model parameters and $\mathcal{T}$ is the target domains. $\mathcal{L}_{\text{task}}$ denotes the task-specific loss function, which is the pixel-wise cross-entropy loss in this paper. $D(\mathcal{S}, \mathcal{T})$ denotes the distribution distance between the source domain and target domains in semantic space. It is constrained to be lower than $\rho$ for semantic consistency. Inspired by Eq. 1, we propose a novel approach to generate a dynamic source domain $\mathcal{S}^+$, which can help us to reduce the domain shifts between the synthetic domain $\mathcal{S}$ and real-world domains $\mathcal{T}$ during training.

### 3.1 OVERVIEW

In the introduction, we show that the image style is an important factor that influences the DG performance. In addition, the channel-wise mean and standard deviation of an image, which is called

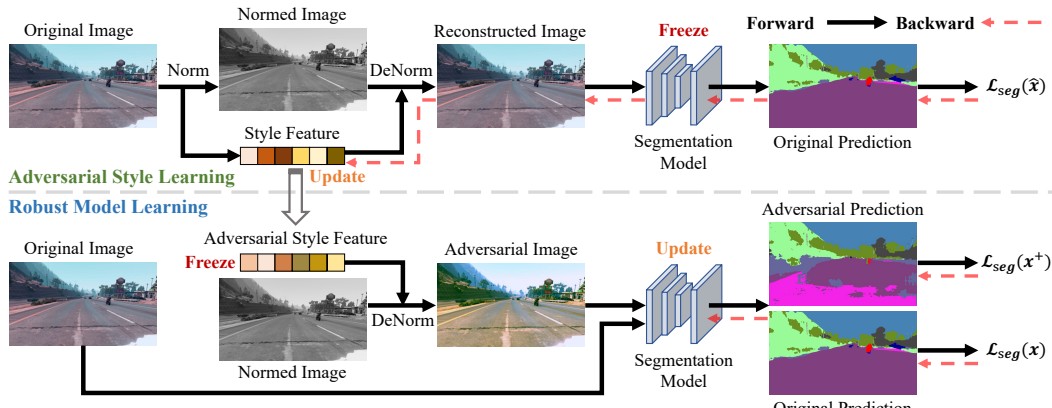

Figure 3: The framework of the proposed adversarial style augmentation.

style feature, can well represent the image style. Inspired by that, we propose the adversarial style augmentation approach (**AdvStyle**) for domain generalized semantic segmentation. AdvStyle can dynamically generate images with new styles during training and effectively improve the generalization ability of the segmentation model. AdvStyle includes two steps: adversarial style learning and robust model training. In adversarial style learning, we first decompose the image into normalized image and style feature and then update the style feature by adversarial segmentation loss. In robust model training, we compose the adversarial image by the normalized image and learned adversarial style feature. The model is then optimized with both original and adversarial images. These two steps are implemented in each training iteration, enabling us dynamically generate hard stylized samples for the current model. Our overall framework is illustrated in Fig .3.

## 3.2 ADVERSARIAL STYLE LEARNING

Given an training image $x$ at each iteration, we first compute the channel-wise mean $\mu$ and standard deviation $\sigma$ of $x$ and then obtain the normalized image $\bar{x}$ by normalization:

$$\mu = \frac{1}{HW} \sum_{h \in H, w \in W} x_{h,w}, \quad \sigma = \sqrt{\frac{1}{HW} \sum_{h \in H, w \in W} (x_{h,w} - \mu)^2}, \quad \bar{x} = \frac{x - \mu}{\sigma}, \quad (2)$$

where $H$ and $W$ denote the spatial size of $x$.

After that, we initialize the adversarial style feature $\mu^+$ and $\sigma^+$ by $\mu$ and $\sigma$, which are regarded as learnable parameters. Then we reconstruct the image with $\mu^+$, $\sigma^+$ and $\bar{x}$ and forward it into the network for loss computation. During the backward, the parameters of the network are fixed and the adversarial style feature is updated by:

$$\mu^+ \leftarrow \mu^+ + \gamma \nabla_{\mu^+} \mathcal{L}_{\text{seg}}(\theta; \hat{x}), \quad \sigma^+ \leftarrow \sigma^+ + \gamma \nabla_{\sigma^+} \mathcal{L}_{\text{seg}}(\theta; \hat{x}), \quad (3)$$

where $\gamma$ is the learning rate for adversarial style learning and $\mathcal{L}_{\text{seg}}$ is the cross-entropy loss. $\hat{x} = \bar{x} \cdot \sigma^+ + \mu^+$ is the reconstructed image. Notice that the style feature is optimized by the adversarial gradient of $\mathcal{L}_{\text{seg}}$. Indeed, the adversarial style learning process can be iterated multiple times. In our experiment, we find that one-step adversarial style learning achieves similar results with multi-step ones but is much more efficient. Therefore, we only update the style feature once.

## 3.3 ROBUST MODEL TRAINING

Given the learned adversarial style feature ($\mu^+$ and $\sigma^+$), we use it to generate the adversarial sample $x^+$ with the corresponding normalized image:

$$x^+ = \bar{x} \cdot \sigma^+ + \mu^+. \quad (4)$$

Then the original image $x$ and the generated adversarial image $x^+$ are forwarded to the model for optimization, which can be formulated by,

$$\min_{\theta} \mathcal{L}_{\text{seg}}(\theta; x) + \mathcal{L}_{\text{seg}}(\theta; x^+). \quad (5)$$

The detailed training procedure and Pytorch-like pseudo-code can be found in Appendix B. During testing, we directly input the original samples into the network without implementing AdvStyle.

## 3.4 DISCUSSION

Adversarial data augmentation have been studied by several works for DG in image classification. Most of them (Qiao et al., 2020; Volpi et al., 2018) generate pixel-wise perturbations on the training image $x$, which usually require additional constraint loss to guarantee the semantic consistency in Eq. 1. In addition, these works focus on the image classification task where the recognition result is mostly related to global feature. However, in semantic segmentation, the model needs to produce the per-pixel predictions so that it is more difficult to ensure the pixel-wise semantic consistency during adversarial learning. Instead, our AdvStyle varies the style feature of the image, which will maintain the semantic content of most pixels and thus can well guarantee the pixel-wise semantic consistency for semantic segmentation. We conduct experiments in Table 3, which demonstrate the superiority of the proposed AdvStyle over pixel-wise adversarial learning.

## 4 EXPERIMENTS

### 4.1 EXPERIMENTAL SETUP

**Datasets.** For the synthetic-to-real domain generalization (DG), we use one of the synthetic datasets (GTAV (Richter et al., 2016) or SYNTHIA (Ros et al., 2016)) as the source domain and evaluate the model performance on three real-world datasets (CityScapes (Cordts et al., 2016), BDD-100K (Yu et al., 2020), and Mapillary (Neuhold et al., 2017)). GTAV (Richter et al., 2016) contains 24,966 images with the size of 1914×1052. It is splited into 12,403, 6,382, and 6,181 images for training, validating, and testing. SYNTHIA (Ros et al., 2016) contains 9,400 images of 960×720, where 6,580 images are used for training. We use the validation sets of the three real-world datasets for evaluation. CityScapes (Cordts et al., 2016) contains 500 validation images of 2048×1024, collected primarily in Germany. BDD-100K (Yu et al., 2020) and Mapillary (Neuhold et al., 2017) contain 1,000 validation images of 1280×720 and 2,000 validation images of 1920×1080, respectively.

**Implementation Details.** Following Choi et al. (2021), we use DeepLabV3+ (Chen et al., 2018b) as the segmentation model. The segmentation model is constructed by three backbones, including MobileNetV2 (Sandler et al., 2018), ResNet-50 (He et al., 2016) and ResNet-101. We adopt SGD optimizer with an initial learning rate 0.01, momentum 0.9 and weight decay $5×10^{-4}$ to optimize the model. The polynomial decay (Liu et al., 2015) with the power of 0.9 is used as the learning rate scheduler. The learning rate of AdvStyle $\gamma$ is set to 3. All models are trained for 40K iterations with a batch size of 16. Four widely used data augmentation techniques are used during training, including color jittering, Gaussian blur, random cropping and random flipping. The input image is randomly cropped to 768×768 for training. The image of the original size is used for testing.

**Evaluation Metric.** Following Choi et al. (2021), the model obtained by the last training iteration is used to evaluate the mIoU performance on the three real-world validation sets. For each method, we report the result averaged on 3 runs. When using GTAV as the source domain, we use the 19 shared semantic categories for training and evaluation. When using SYNTHIA as the source domain, we use 16 shared categories for training and evaluation, *i.e.,* ignoring the train, truck, and terrain categories.

### 4.2 EVALUATION

**Effectiveness of AdvStyle on Different Models.** Our AdvStyle is a model-agnostic method, which can be directly applied to different models without modifying the models. To verify the effectiveness of AdvStyle, we apply it to models with different backbones and normalization modules. The backbones include MobileNetV2, ResNet-50 and ResNet-101. The normalization modules include vanilla batch norm (baseline), instance-batch norm (IBN-Net (Pan et al., 2018)), and instance selective whitening (ISW (Choi et al., 2021)). In Table 1, we show the results of using GTAV as the source domain. We can make the following conclusions. First, injecting instance-batch norm (IBN-Net) and instance selective whitening (ISW) modules can consistently improve the performance of the baseline. Second, the proposed AdvStyle can significantly enhance the generalization performance of the baseline model for all backbones. Specifically, the average mIoU is increased from 26.03%, 27.42% and 31.47% to 32.11%, 37.39% and 37.34% for MobileNetV2, ResNet-50 and ResNet-101, respectively. In addition, the baseline with AdvStyle produces higher average mIoU than IBN-Net

| Methods | MobileNetV2 | | | | ResNet-50 | | | | ResNet-101 | | | |
|---|---|---|---|---|---|---|---|---|---|---|---|---|
| GTAV→ | C | B | M | Mean | C | B | M | Mean | C | B | M | Mean |
| Baseline | 25.92 | 25.73 | 26.45 | 26.03 | 28.95 | 25.14 | 28.18 | 27.42 | 32.97 | 30.77 | 30.68 | 31.47 |
| +AdvStyle | **31.81** | **33.01** | **31.50** | **32.11** | **39.62** | **35.54** | **37.00** | **37.39** | **39.52** | **36.39** | **36.10** | **37.34** |
| IBN-Net | 30.14 | 27.66 | 27.07 | 28.29 | 33.85 | 32.30 | 37.75 | 34.63 | 37.37 | 34.21 | 36.81 | 36.13 |
| +AdvStyle | **32.45** | **31.55** | **33.09** | **32.36** | **39.32** | **36.42** | **40.82** | **38.85** | **44.04** | **39.96** | **42.67** | **42.22** |
| ISW | 30.86 | 30.05 | 30.67 | 30.53 | 36.58 | 35.20 | 40.33 | 37.37 | 37.20 | 33.36 | 35.57 | 35.38 |
| +AdvStyle | **33.23** | **31.84** | **32.00** | **32.36** | **39.60** | **38.59** | **41.89** | **40.03** | **43.44** | **40.32** | **41.96** | **41.91** |

Table 1: Evaluation of the proposed AdvStyle on different methods (Baseline, IBN-Net (Pan et al., 2018) and ISW(Choi et al., 2021)) and backbones (MobileNetV2 (Sandler et al., 2018), ResNet-50 (He et al., 2016), and ResNet-101). All models are trained on the GTAV training set and tested on CityScapes (C), BDD-100K (B), and Mapillary (M) validation sets.

| Methods (SYNTHIA→) | CityScapes | BDD | Mapillary | Mean |
|---|---|---|---|---|
| Baseline (Choi et al., 2021) | 34.94 | 21.96 | 27.94 | 28.28 |
| **Baseline+AdvStyle** | **37.59** | **27.45** | **31.76** | **32.27** |
| IBN-Net (Pan et al., 2018) | 35.83 | 23.62 | 28.88 | 29.44 |
| **IBN-Net+AdvStyle** | **38.72** | **28.55** | **33.59** | **33.62** |
| ISW (Choi et al., 2021) | 35.27 | 23.54 | 26.72 | 28.51 |
| **ISW+AdvStyle** | **39.74** | **28.33** | **32.87** | **33.65** |

Table 2: Results of using SYNTHIA as the source domain. The backbone is ResNet-101.

| Color Jittering | Gaussian Blur | AdvPixel | AdvStyle | CityScapes | BDD | Mapillary | Mean |
|---|---|---|---|---|---|---|---|
| - | - | - | - | 21.64 | 22.85 | 24.22 | 22.91 |
| ✓ | - | - | - | 26.36 | 23.82 | 26.33 | 25.50 |
| - | ✓ | - | - | 25.77 | 24.05 | 26.71 | 25.51 |
| - | - | ✓ | - | 23.34 | 28.42 | 30.64 | 27.46 |
| - | - | - | ✓ | **37.51** | **33.74** | **34.73** | **35.32** |
| ✓ | ✓ | - | - | 28.95 | 25.14 | 28.18 | 27.42 |
| ✓ | ✓ | ✓ | - | 35.42 | 33.28 | 33.23 | 33.97 |
| ✓ | ✓ | - | ✓ | **39.62** | **35.54** | **37.00** | **37.39** |

Table 3: Comparison of different augmentations. Source: GTAV; Backbone: ResNet-50.

and ISW for all backbones. Third, when adding AdvStyle, the results of IBN-Net and ISW can be further improved for all settings. For example, when using ResNet-101 as the backbone, AdvStyle improves the average mIoU of IBN-Net and ISW by 6.09% and 6.53%, respectively. In Table 2, we show the results of using SYNTHIA as the source domain and also observe clear improvements for AdvStyle. These results verify the prominent advantage of our AdvStyle on different models.

**Comparison of Different Augmentation Techniques.** We investigate the impact of different augmentation methods, including color jittering, Gaussian blur, AdvPixel (Volpi et al., 2018) and the proposed AdvStyle. AdvPixel is a state-of-the-art method for domain generalized image classification. The main difference between AdvPixel and AdvStyle is that AdvPixel learns pixel-wise adversarial example while AdvStyle learns style-wise adversarial example. We reproduce AdvPixel in our setting and select the adversarial learning rate (=10) that achieves the best performance. The random cropping and random flipping are used in default.

Results in Table 3 show that all four augmentation methods can improve the generalization performance. Importantly, our AdvStyle produces significant improvement compared to other three methods. Specifically, when using AdvStyle, the average mIoU is increased from 22.91% to 35.32%. This improvement is about 8% higher than the other 3 methods. Moreover, AdvStyle is well complementary to color jittering and Gaussian blur. When combining these three methods, the mIoU is further improved in all target domains. Compared to AdvPixel, our AdvStyle achieves clearly higher performance, no matter using color jittering and Gaussian blur. This demonstrates the advantage of learning adversarial style in domain generalized semantic segmentation.

| Methods (GTAV) | CityScapes | BDD | Mapillary | Mean |
|---|---|---|---|---|
| Baseline | 28.95 | 25.14 | 28.18 | 27.42 |
| RandStyle | 33.40 | 34.14 | 31.67 | 33.07 |
| MixStyle | 35.53 | 32.41 | 35.87 | 34.60 |
| CrossStyle | 37.26 | 32.40 | 34.09 | 34.58 |
| **AdvStyle** | **39.62** | **35.54** | **37.00** | **37.39** |

Table 4: Comparison of different style-aware methods. Source: GTAV; Backbone: ResNet-50.

**Comparison of Different Style-Aware Methods.** In Table 4, we compare AdvStyle with three style-aware augmentation methods, including MixStyle (Zhou et al., 2021), CrossStyle (Tang et al., 2021) and RandStyle. MixStyle mixes the styles of two samples with a convex weight while CrossStyle directly swaps the styles of two samples. RandStyle can be regarded a reduction of our AdvStyle, which randomly adds Gaussian noise into the style feature. All style-aware methods are implemented on the image-level for fair comparison. We can find that (1) all style-aware methods can consistently improve the performance on all target domains and (2) AdvStyle achieves the best results. The first finding verifies the effectiveness of augmenting image styles and the second finding shows the benefit of learning adversarial styles over other style-aware methods for domain generalized semantic segmentation.

## 4.3 COMPARISON WITH STATE-OF-THE-ART METHODS

In Table 5, we compare our method with state-of-the-art DG methods in semantic segmentation, including IBN-Net (Pan et al., 2018), SW (Pan et al., 2019), IterNorm (Huang et al., 2019), ISW (Choi et al., 2021), DPRC (Yue et al., 2019) and FSDR (Huang et al., 2021). The source domain is GTAV and the backbones are ResNet-50 and ResNet-101. Note that, since different methods use different segmentation networks (*e.g.*, DeepLabV2 (Chen et al., 2018a), DeepLabV3+ (Chen et al., 2018b) and FCN (Long et al., 2015)), different training sets (*e.g.*, the whole GTAV and the training set of GTAV), different training strategies (*e.g.*, learning rate and optimizer) , different auxiliary data (*e.g.*, ImageNet samples) and different evaluation manners (*e.g.*, the best model and the last model), it is hard to compare them in an absolutely fair way. We show the results of each method as well as the absolute gain against the corresponding baseline.

From Table 5, we can make the following conclusions. **First**, when using the same baseline model, adding AdvStyle can produce better results than IBN-Net, SW, IterNorm and ISW. Moreover, when applying AdvStyle to IBN-Net or ISW, we achieve new state-of-the-art performance for both ResNet-50 and ResNet-101. **Second**, when compared across baselines, "Baseline+AdvStyle" achieves the state-of-the-art mIoU for ResNet-50. On the other hand, when using ResNet-101 as the backbone, "IBN-Net+AdvStyle" produces higher results than DRPC and comparable results with FSDR. Importantly, both FSDR and DRPC use extra ImageNet images and select the best training checkpoints for each target domain. Instead, "IBN-Net+AdvStyle" only utilizes the source data and uses the last training checkpoint to evaluate all target domains. **Third**, when using the best checkpoint for evaluation, we ("ISW+AdvStyle") produce better performance than DRPC and FSDR, leading to the new state-of-the-art results under the "best checkpoint setting". *Even so, we argue that it is more reasonable to use the last checkpoint for evaluating all target domains. This is because we can not always have the right labeled validation sets to select the best model for unseen domains in practice.*

To make more fair comparisons with the state-of-the-art methods, we also use the whole set of GTAV for training ISW and "ISW+AdvStyle". The results with ResNet-101 are reported in the last two rows of Table 5. We can find that using the whole set of GTAV can produce higher results on the CityScapes and the Mapillary datasets.

## 4.4 VISUALIZATION

**Qualitative Comparison of Segmentation Results.** In Fig. 4, we compare the segmentation results for different methods on target domains. It is clear that, the proposed AdvStyle can consistently improve the semgentation results for baseline, IBN and ISW models, especially for the easily-confused classes, *e.g.*, road *vs* sidewalk and building *vs* sky. More results can be found in Appendix E.

| Net | ID | Methods (GTAV) | CityScapes | | BDD | | Mapillary | | Mean | |
|---|---|---|---|---|---|---|---|---|---|---|
| ResNet-50 | I | Baseline | 22.20 | - | N/A | | N/A | | N/A | |
| | I | IBN-Net | 29.60 | 7.40↑ | | | | | | |
| | II | Baseline* | 32.45 | - | 26.73 | - | 25.66 | - | 28.28 | - |
| | II | DRPC§* | 37.42 | 4.97↑ | 32.14 | 5.41↑ | 34.12 | 8.46↑ | 34.56 | 6.28↑ |
| | III | Baseline | 28.95 | - | 25.14 | - | 28.18 | - | 27.42 | - |
| | III | **Baseline+AdvStyle** | **39.62** | 10.67↑ | 35.54 | 10.4↑ | 37.00 | 8.82↑ | 37.39 | 9.97↑ |
| | III | SW | 29.91 | 0.96↑ | 27.48 | 2.34↑ | 29.71 | 1.53↑ | 29.03 | 1.61↑ |
| | III | IterNorm | 31.81 | 2.86↑ | 32.70 | 7.56↑ | 33.88 | 5.7↑ | 32.79 | 5.37↑ |
| | III | IBN-Net | 33.85 | 4.90↑ | 32.30 | 7.16↑ | 37.75 | 9.57↑ | 34.63 | 7.21↑ |
| | III | **IBN-Net+AdvStyle** | 39.32 | 10.37↑ | 36.42 | 11.28↑ | 40.82 | 12.64↑ | 38.85 | 11.43↑ |
| | III | ISW | 36.58 | 7.63↑ | 35.20 | 10.06↑ | 40.33 | 12.15↑ | 37.37 | 9.95↑ |
| | III | **ISW+AdvStyle** | 39.60 | 10.65↑ | **38.59** | 13.45↑ | **41.89** | 13.71 ↑ | **40.03** | 12.61↑ |
| ResNet-101 | I | Baseline* | 33.4 | - | 27.3 | - | 27.9 | - | 29.53 | - |
| | I | IBN-Net* | 40.3 | 6.9↑ | 35.6 | 8.3↑ | 35.9 | 8.0↑ | 37.26 | 7.73↑ |
| | I | FSDR§* | 44.8 | 11.4↑ | 41.2 | 13.9↑ | 43.4 | 15.5↑ | 43.13 | 13.6↑ |
| | II | Baseline* | 33.56 | - | 27.76 | - | 28.33 | - | 29.88 | - |
| | II | DRPC§* | 42.53 | 8.97↑ | 38.72 | 10.96↑ | 38.05 | 9.72↑ | 39.76 | 9.88↑ |
| | III | Baseline | 32.97 | - | 30.77 | - | 30.68 | - | 31.47 | - |
| | III | **Baseline+AdvStyle** | 39.52 | 6.55↑ | 36.39 | 5.62↑ | 36.10 | 5.42↑ | 37.34 | 5.87↑ |
| | III | IBN-Net | 37.37 | 4.40↑ | 34.21 | 3.44↑ | 36.81 | 6.13↑ | 36.13 | 4.66↑ |
| | III | **IBN-Net+AdvStyle** | 44.04 | 11.07↑ | 39.96 | 9.19↑ | 42.67 | 11.99↑ | 42.22 | 10.75↑ |
| | III | ISW | 37.20 | 4.23↑ | 33.36 | 2.59↑ | 35.57 | 4.89↑ | 35.38 | 3.91↑ |
| | III | **ISW+AdvStyle** | 43.44 | 10.47↑ | 40.32 | 9.55↑ | 41.96 | 11.28↑ | 41.91 | 10.44↑ |
| | III | **ISW+AdvStyle*** | **45.62** | 12.65↑ | **41.71** | 10.97↑ | **46.69** | 16.01↑ | **44.67** | 13.20↑ |
| | IV | ISW | 37.51 | - | 33.54 | - | 36.12 | - | 35.72 | - |
| | IV | **ISW+AdvStyle** | **44.51** | - | **39.27** | - | **43.48** | - | **42.42** | - |

Table 5: Comparison with state-of-the-art domain generalization methods. All models use the GTAV as the source domain. For each backbone, models with the same ID are implemented with the same baseline. Models of "ID=I, II and IV" use the whole set (24,966) for training while models of "ID=III" only use the training set (12,403). The absolute gain of each model is calculated over the corresponding baseline. § denotes extra using the ImageNet images. * indicates using the best trained checkpoints for evaluating each target domain.

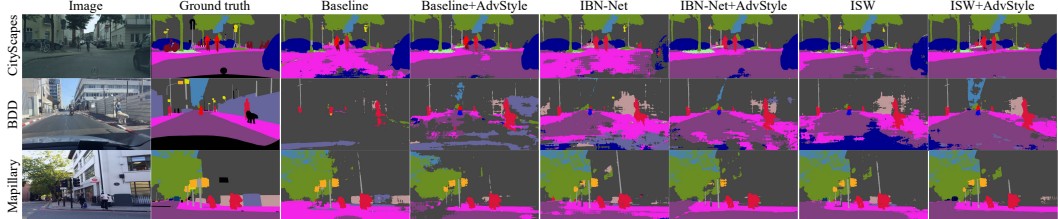

Figure 4: Qualitative comparison of segmentation results. Source: GTAV; Backbone: ResNet-50.

**t-SNE of Styles.** In Fig. 5, we visualize the style features generated by AdvStyle during the training phase for the GTAV training set, where ResNet-50 is used as the backbone. We can find that AdvStyle can continuously generate new style features that are different from the original distribution. The new style features have the chance to be located at the distributions of other datasets during the training process. Moreover, AdvStyle will also generate styles that are out of the distributions of the four datasets (G, C, B, M) and may appear in other unseen domains. The visualization results further demonstrate that AdvStyle can encourage the model to meet more diverse and unseen styles during training, leading to a more robust model.

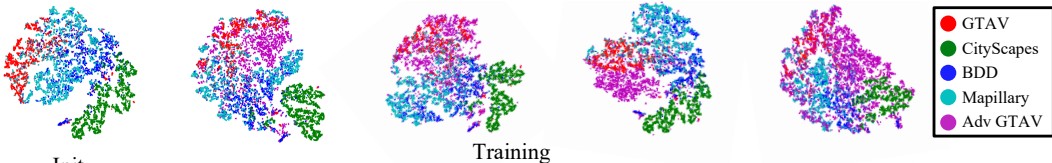

Figure 5: t-SNE visualization of adversarial style features during training.

| Method | SVHN | MNIST-M | SYN | USPS | Avg. |
|---|---|---|---|---|---|
| ERM | 27.8 | 52.7 | 39.7 | 76.9 | 49.3 |
| CCSA | 25.9 | 49.3 | 37.3 | 83.7 | 49.1 |
| d-SNE | 26.2 | 51.0 | 37.8 | **93.2** | 52.1 |
| JiGen | 33.8 | 57.8 | 43.8 | 77.2 | 53.1 |
| ADA | 35.5 | 60.4 | 45.3 | 77.3 | 54.6 |
| M-ADA | 42.6 | 67.9 | 49.0 | 78.5 | 59.5 |
| ME-ADA | 42.6 | 63.3 | 50.4 | 81.0 | 59.3 |
| **ERM+AdvStyle** | **50.4** | **73.4** | **58.7** | 81.6 | **66.0** |

Table 6: Accuracy of single domain generalization on Digits. MNIST is used as the training set, and the results on different testing domains are reported in different columns.

| Method | Art. | Car. | Ske. | Pho. | Avg. |
|---|---|---|---|---|---|
| ERM | 67.4 | 74.4 | 51.4 | 42.6 | 58.9 |
| JiGen | 69.1 | 74.6 | 52.4 | 41.5 | 59.4 |
| RSC | 68.8 | 74.5 | 53.6 | 41.9 | 59.7 |
| L2D | 74.3 | 77.5 | 54.4 | 45.9 | 63.0 |
| **ERM+AdvStyle** | **75.8** | 76.6 | 58.1 | 51.1 | 65.4 |
| **RSC+AdvStyle** | 75.1 | **78.0** | **58.9** | **55.5** | **66.8** |

Table 7: Accuracy of single domain generalization on PACS. One domain (name in column) is used as the training (source) data and the other domains are used as the testing (target) data.

### 4.5 EVALUATION ON IMAGE CLASSIFICATION TASK

To verify the versatility of the proposed AdvStyle, we evaluate it on single DG in image classification. Experiments are conducted on two popular DG datasets, *i.e.*, Digits and PACS. The details of the datasets and implementation can be found in Appendix D.

**Results on Digits.** In Table 6, we compare with the baseline (ERM (Vapnik, 2013)) and 6 state-of-the-art methods, including CCSA (Motiian et al., 2017), d-SNE (Xu et al., 2019), JiGen (Carlucci et al., 2019), ADA (Volpi et al., 2018), M-ADA (Qiao et al., 2020) and ME-ADA (Zhao et al., 2020). For AdvStyle, we implement it with ERM. It is clear that, our AdvStyle can significantly improve the accuracy of ERM on all target domains. In addition, the proposed AdvStyle outperforms the other state-of-the-art methods by a large margin. For example, AdvStyle is higher than the best competitor (ME-ADA (Zhao et al., 2020)) by 6.7% for the accuracy averaged over 4 target domains.

**Results on PACS.** We compare with the baseline (ERM (Vapnik, 2013)) and three state-of-the-art DG methods, including JiGen (Carlucci et al., 2019), RSC (Huang et al., 2020) and L2D (Wang et al., 2021). We reproduce JiGen, RSC and L2D with their official source codes. All methods use the same baseline (ERM). Results in Table 7 show that JiGen and RSC produce limited improvements. Instead, our AdvStyle can significantly increase the accuracy on all domains for both ERM and RSC. Compared to the recent published work (L2D), our method (ERM+AdvStyle) outperforms it by 2.4% in average accuracy.

The results on Digits and PACS demonstrate that our AdvStyle can also be effectively applied to single domain generalized image classification and can achieve state-of-the-art accuracy.

## 5 CONCLUSION

In this paper, we propose a novel augmentation approach, called adversarial style augmentation (**AdvStyle**), for domain generalization (DG) in semantic segmentation. AdvStyle dynamically generates hard stylized images by learning adversarial image-level style feature, which can encourage the model learning with more diverse samples. With AdvStyle, the model can refrain from the problem of overfitting on the source domain and thus can be more robust to the style variations of unseen domains. AdvStyle is easy to implement and can be directly integrated with different models without modifying the network structures and learning strategies. Experiments on two synthetic-to-real settings show that AdvStyle can largely improve the generalization performance and achieve state-of-the-art performance. In addition, AdvStyle can be employed to single DG in image classification and obtain significant improvement.

ETHICS STATEMENT

This paper presents an effective approach for synthetic-to-real domain generalization (DG) in semantic segmentation, which can help to improve safety in autonomous driving. We do not find obvious Ethics issues for our approach since we only use synthetic data for model training. Be that as it may, one potential issue is that the performance of existing DG approaches (including our approach) in semantic segmentation is still far away from the practical demand, especially when encountering extreme conditions. This may lead the drivers not to be completely at ease when using the autonomous driving system.

REPRODUCIBILITY STATEMENT

We provide the implementation details and dataset description in Sec. 4.1 and Appendix D for semantic segmentation and image classification, respectively. In addition, the algorithm and Pytorch-like pseudo-code are presented in Appendix B, which allow the researchers to easily reproduce / integrate our method with existing semantic segmentation and image classification approaches.

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

## A  PARAMETER ANALYSIS

The proposed AdvStyle has one parameter, *i.e.*, the adversarial learning rate $\gamma$. To study the impact of $\gamma$, we vary it in the range of $[0.1, 40]$. Results in Fig. 6 show that AdvStyle can significantly improve the performance on all target domains even with a small value of $\gamma$. The best results are achieved when $\gamma$ is between 1 and 10. Assigning a too large value to $\gamma$ (*e.g.*, 40) may produce unrealistic styles and thus hampers the model training.

## B  ALGORITHM AND PYTORCH-LIKE PSEUDO-CODE

The training procedure and Pytorch-like pseudo-code are shown in Alg. 1 and Fig. 7, respectively.

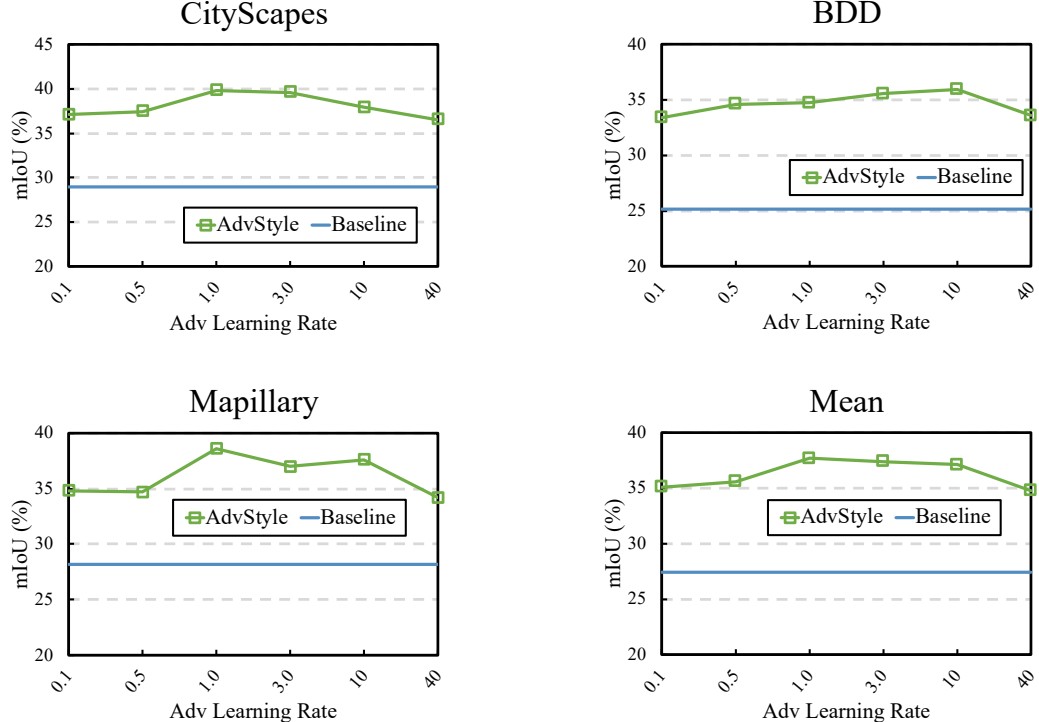

Figure 6: Influence of the adversarial learning rate.

---

**Algorithm 1** The training procedure of AdvStyle.

---

**Inputs:** labeled source domain $\mathcal{S}$, segmentation model $\mathcal{F}$ parameterized by $\theta$, batch size $N_b$, total training iterations $max\_iter$, adversarial learning rate $\gamma$, and model learning rate $\alpha$.

**Outputs:** Optimized model $\mathcal{F}$ parameterized with $\theta$.

1: **for** $i$ in $max\_iter$ **do**
2:     Sample mini-batch $\mathcal{X}$ with $N_b$ images;
3:     // Stage 1: Adversarial Style Learning.
4:     Compute channel-wise mean $\mu$, standard deviation $\sigma$ and normalized images $\bar{\mathcal{X}}$ with Eq. 2;
5:     Initialize adversarial style feature: $\mu^+ \leftarrow \mu$, $\sigma^+ \leftarrow \sigma$;
6:     Compute adversarial segmentation loss $-\mathcal{L}_{\text{seg}}$;
7:     Optimize $\mu^+$ and $\sigma^+$ with Eq. 3;
8:     // Stage 2: Robust Model Training.
9:     Generate adversarial images $\mathcal{X}^+$ with $\bar{\mathcal{X}}$, $\mu^+$ and $\sigma^+$ by Eq. 4;
10:     Compute the overall training loss $\mathcal{L}_{\text{seg}}(\theta; \mathcal{X}) + \mathcal{L}_{\text{seg}}(\theta; \mathcal{X}^+)$ by Eq 5;
11:     Optimize the segmentation model $\mathcal{F}$: $\theta \leftarrow \theta - \alpha\nabla_\theta \left(\mathcal{L}_{\text{seg}}(\theta; \mathcal{X}) + \mathcal{L}_{\text{seg}}(\theta; \mathcal{X}^+)\right)$;
12: **end for**
13: **Return** $\mathcal{F}$ parameterized with $\theta$.

---

## C    RESULTS OF MULTI-SOURCE SETTING

In Table 8, we evaluate the models under the multi-source domain generalization setting, where both GTAV and SYNTHIA are used as the source data. The compared methods include baseline (Choi et al., 2021), IBN-Net (Pan et al., 2018), ISW (Choi et al., 2021) and our AdvStyle. Clearly, AdvStyle consistently improves the results of ISW, further verifying the effectiveness of the proposed AdvStyle.

```python
1   import torch
2
3   def AdvStyle(input, gt, net, optim, adv_lr):
4       ```
5       Args:
6               input: source images
7               gt: ground-truth labels
8               net: segmentation network
9               optim: optimizer of net
10              adv_lr: learning rate of AdvStyle
11      ```
12      ### Adversarial Style Learning
13
14      # Get style feature and normalized image
15      B = input.size(0)
16      mu = input.mean(dim=[2, 3], keepdim=True)
17      var = input.var(dim=[2, 3], keepdim=True)
18      sig = (var + 1e-5).sqrt()
19      mu, sig = mu.detach(), sig.detach()
20      input_normed = (input - mu) / sig
21      input_normed = input_normed.detach().clone()
22
23      # Set learnable style feature and adv optimizer
24      adv_mu, adv_sig = mu, sig
25      adv_mu.requires_grad_(True)
26      adv_sig.requires_grad_(True)
27      adv_optim = torch.optim.SGD(params=[adv_mu, adv_sig], lr=adv_lr, momentum=0, weight_decay=0)
28
29      # Optimize adversarial style feature
30      adv_optim.zero_grad()
31      adv_input = input_normed * adv_sig+ adv_mu
32      adv_output = net(adv_input)
33      adv_loss = torch.nn.functional.cross_entropy(adv_output, gt)
34      (- adv_loss).backward()
35      adv_optim.step()
36
37      ### Robust Model Training
38      net.train()
39      optim.zero_grad()
40      adv_input = input_normed * adv_sig + adv_mu
41      inputs = torch.cat((input, adv_input), dim=0)
42      gt = torch.cat((gt, gt), dim=0)
43      outputs = net(inputs)
44      loss = F.cross_entropy(outputs, gt)
45      loss.backward()
46      optim.step()
```

Figure 7: The Pytorch-like pseudo-code of AdvStyle.

| Methods (GTAV+SYNTHIA→) | CityScapes | BDD | Mapillary | Mean |
|---|---|---|---|---|
| Baseline (Choi et al., 2021) | 35.46 | 25.09 | 31.94 | 30.83 |
| IBN-Net (Pan et al., 2018) | 35.55 | 32.18 | 38.09 | 35.27 |
| ISW (Choi et al., 2021) | 37.69 | 34.09 | 38.49 | 36.75 |
| **ISW+AdvStyle** | **39.29** | **39.26** | **41.14** | **39.90** |

Table 8: Results of using GTAV and SYNTHIA as the source data. The backbone is ResNet-50.

## D  DETAILS OF SINGLE DOMAIN GENERALIZATION IN IMAGE CLASSIFICATION

**Digits** includes five domains (MNIST (LeCun et al., 1989), SVHN (Netzer et al., 2011), MNIST-M (Ganin & Lempitsky, 2015), SYN (Ganin & Lempitsky, 2015), and USPS (Hull, 1994)) of 10 classes. We use MNIST as the source domain and evaluate the model performance on the other 4 domains. Following ADA (Volpi et al., 2018), we use the ConvNet architecture (LeCun et al., 1989) as the model and use Adam optimizer with learning rate $10^{-4}$ for optimization. The overall training iteration is set to 10,000 with a batch size of 32. We set the learning rate of AdvStyle to 20,000[1].

---

[1]Due to the absent of batch normalization layer, the gradient is very small on the style feature. Therefore, we set a large learning rate for AdvStyle.

**PACS** (Li et al., 2017) contains four domains (Artpaint, Cartoon, Sketch, and Photo) of 7 classes. For evaluation, we select one of them as the source domain and the other domains as the target domains. Following RSC (Huang et al., 2020), we use the ResNet18 (He et al., 2016) pretrained on ImageNet (Deng et al., 2009) as the backbone and add a fully-connected layer as the classification head. We train the model by SGD optimizer. The learning rate is initially set to 0.004 and divided by 10 after 24 epochs. The model is trained for 30 epochs in total with a batch size of 128. The learning rate of AdvStyle is set to 3.

**Baseline.** The baseline model is the vanilla empirical risk minimization (ERM) (Vapnik, 2013), which directly uses the source domain to train the model with classification loss.

## E  MORE VISUALIZATIONS

**Segmentation Results**. In Fig. 8, Fig. 9, and Fig. 10, we provide more segmentation results for the baseline and "baseline+AdvStyle".

**Examples of AdvStyle**. In Fig. 11, we illustrate more examples generated by AdvStyle.

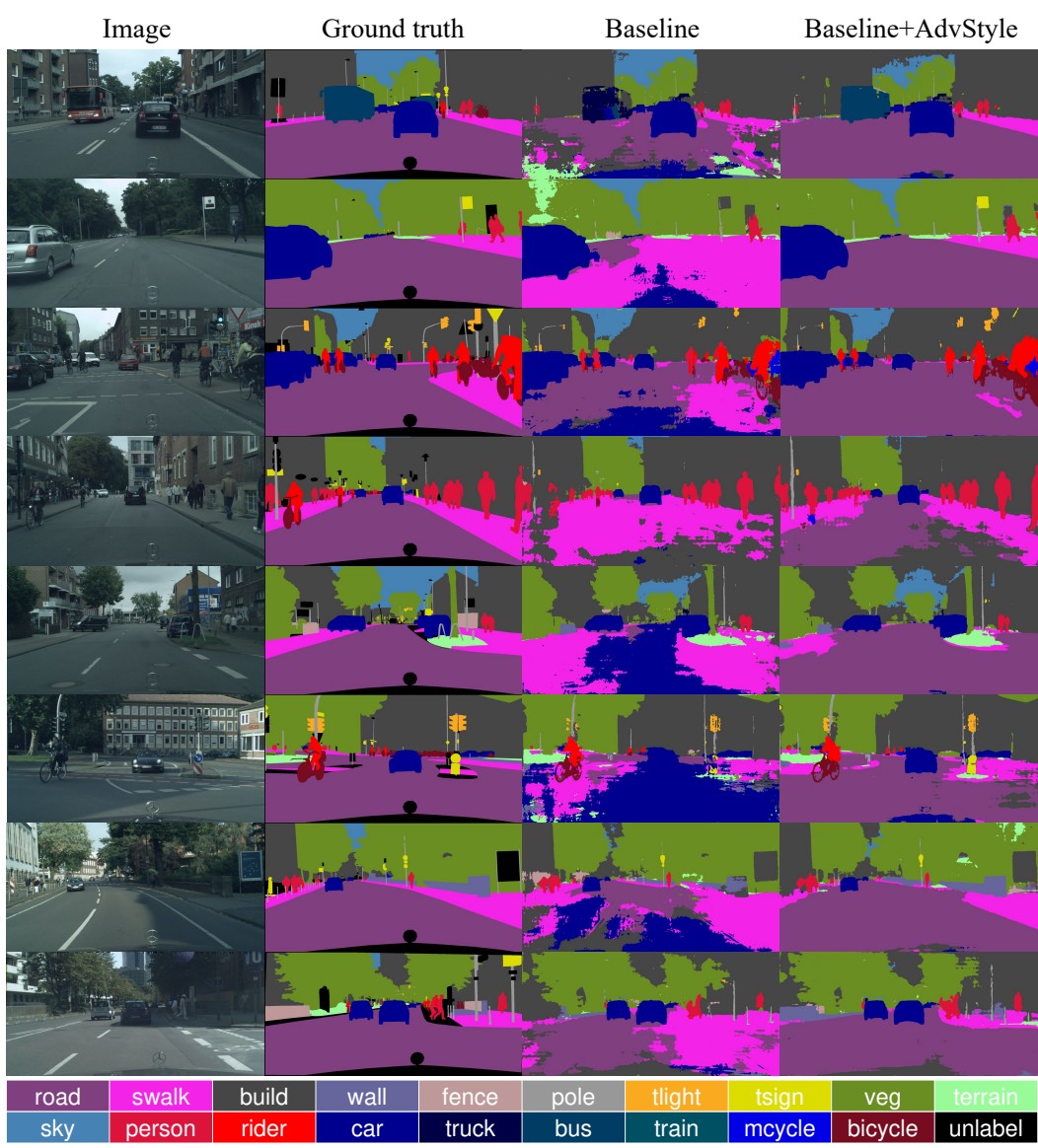

Figure 8: Segmentation results on CityScapes. Source: GTAV; Backbone: ResNet-50.

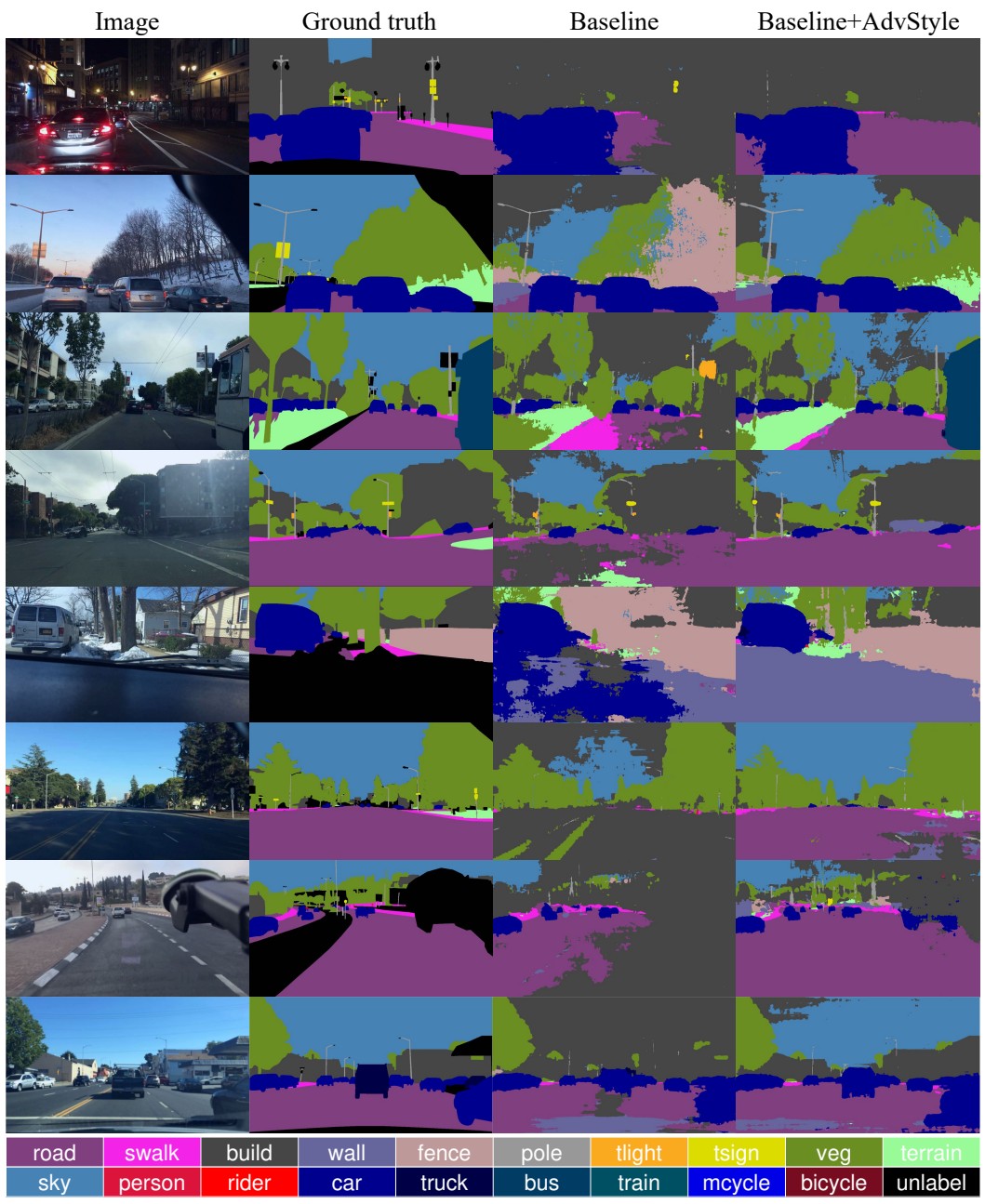

Figure 9: Segmentation results on BDD-100K. Source: GTAV; Backbone: ResNet-50.

Image        Ground truth        Baseline        Baseline+AdvStyle

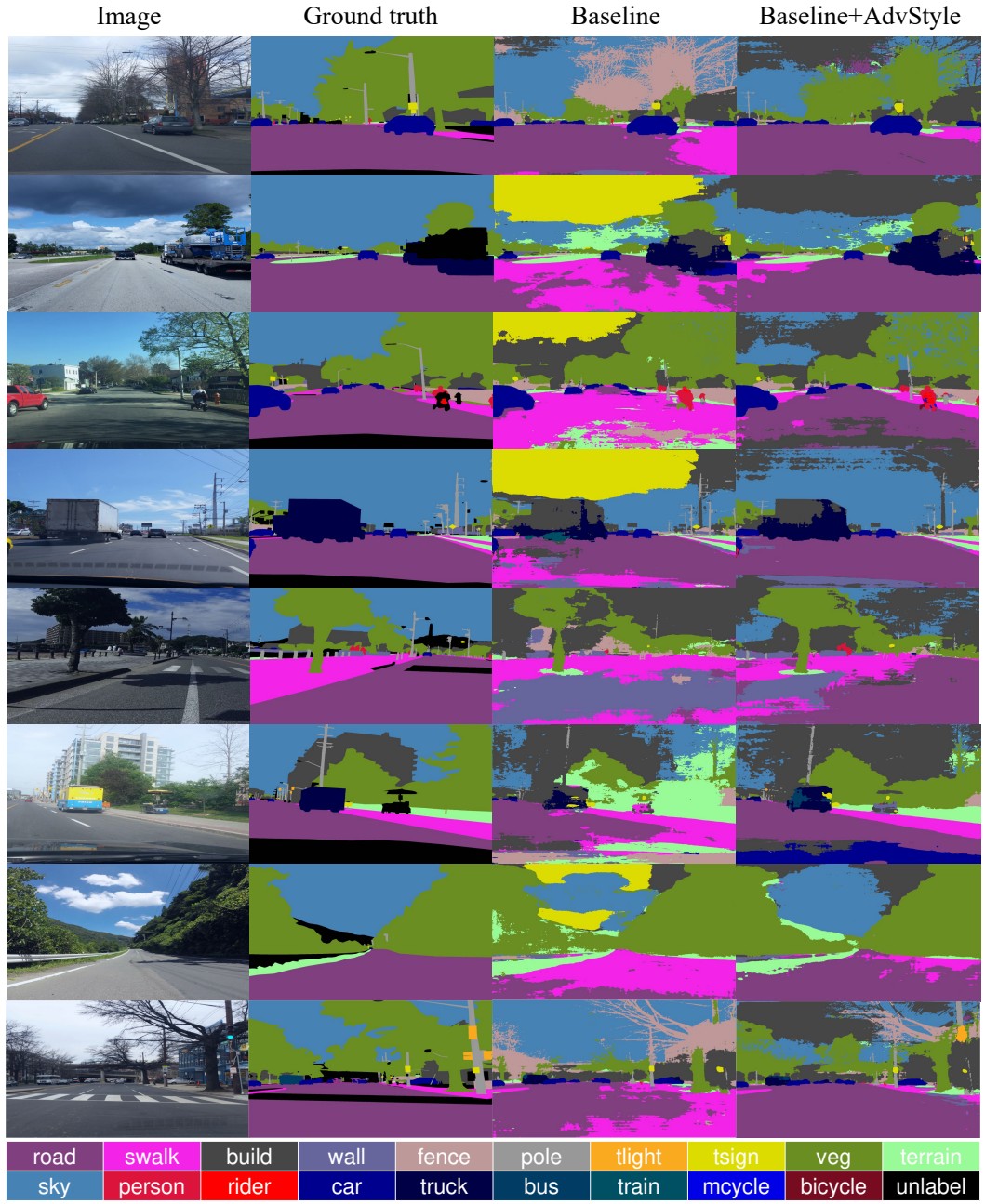

| road | swalk | build | wall | fence | pole | tlight | tsign | veg | terrain |
| sky | person | rider | car | truck | bus | train | mcycle | bicycle | unlabel |

Figure 10: Segmentation results on Mapillary. Source: GTAV; Backbone: ResNet-50.

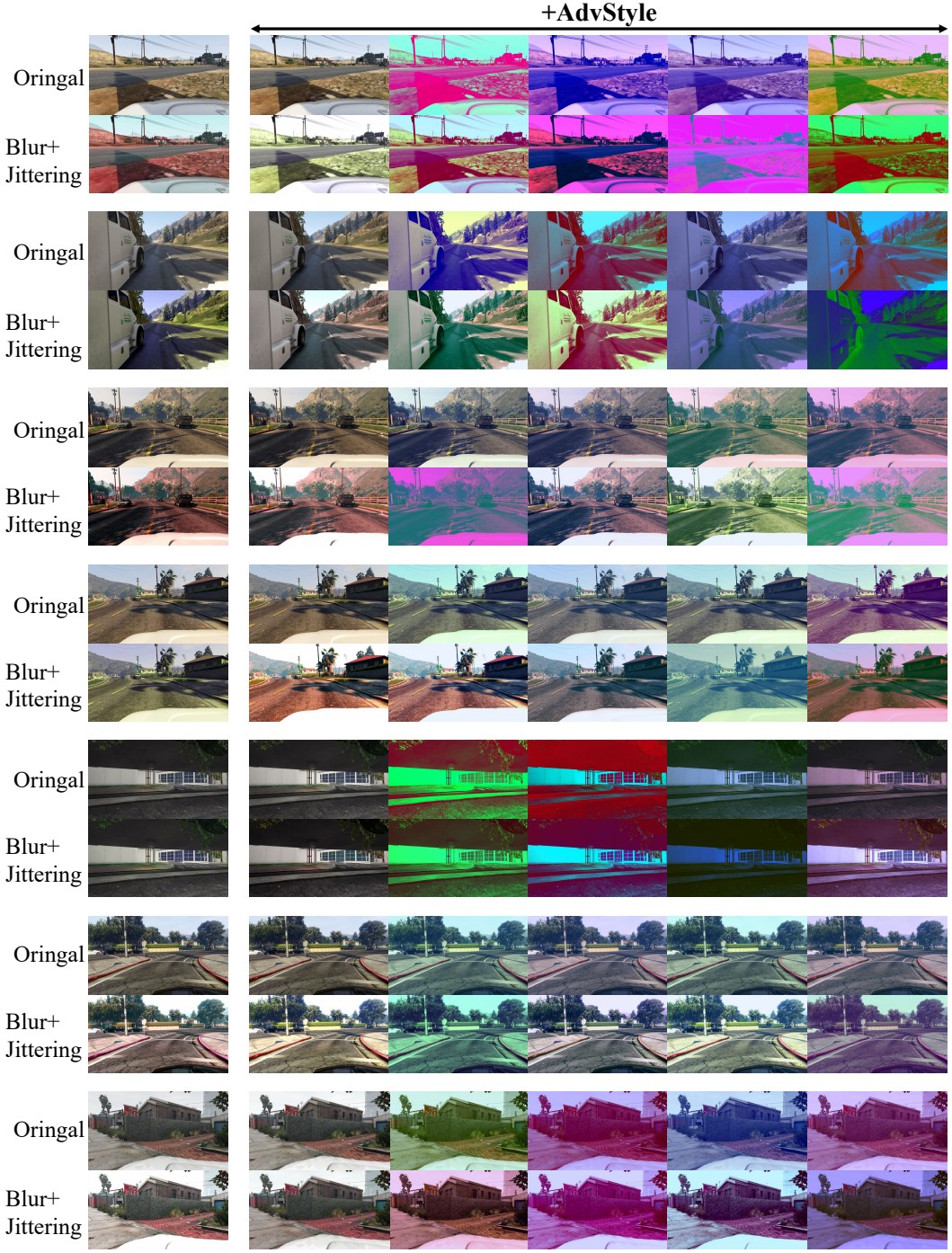

Figure 11: Examples of adversarial style augmentation. Source: GTAV; Backbone: ResNet-50.

