# OpenReview forum: "Adversarial Style Augmentation for Domain Generalized Urban-Scene Segmentation"
_ICLR.cc/2022/Conference — ICLR 2022 Submitted_

### Official Review · Reviewer_38TQ · 2021-10-30

**Correctness:** 4
**Technical Novelty And Significance:** 3
**Empirical Novelty And Significance:** 3
**Recommendation:** 8
**Confidence:** 5

**Main Review:**

Pros:
- Definitely, the main point in favor of this paper is the good results obtained by the approach. I believe the approach is well evaluated, making the right choices of synthetic datasets (training), real datasets (testing), semantic segmentation architectures, and baselines (e.g. ISW, which is one of the best performing approaches for domain generalization in semantic segmentation).

- The approach is orthogonal to other solutions, such as ISW. The combination of ISW and AdvStyle leads to even better accuracy in the downstream task (semantic segmentation).

- The technique is conceptually simple and easy to reproduce.


##########################################################################

Cons:

- The t-SNE visualization (Figure 5) doesn’t seem to convey much information regarding what is happening during the training process. I understand it is typical to visualize features spaces in DA/DG papers, but in this case, it is not clear to me that the figure helps make any point. On the contrary, I would argue it distracts from the main point.

##########################################################################

Questions during rebuttal period:

I don’t have major concerns with this paper. However, I have a recommendation/thought for the authors.
- Why focus just on driving scenes? It should be relatively straightforward to evaluate this approach in other types of scenes, for instance, indoor scenes, by leveraging NYU dataset + SceneNet or InteriorNet. This will help make an even stronger point.

#########################################################################
Miscellanea:

- Figure 1 is cluttered and hard to understand. I would recommend removing b) and d).

- Figure 5, as mentioned above, doesn’t seem to help with the narrative.


**Summary Of The Paper:**

 The paper presents an adversarial style augmentation approach (referred to as AdvStyle) with the objective of improving the generalization of models from a synthetic domain (training) to a real domain (testing), for the task of scene segmentation in the context of driving. The approach is based on the idea of adversarial style learning and the training of a model using both original and adversarial images. A strong set of experiments showcase the benefits of the approach with respect to the state of the art.


**Summary Of The Review:**

My vote is towards accepting this paper. Similar concepts have been applied in the past to improve domain generalization but never reaching the quality achieved by the proposed approach. It is exciting to see how the application of adversarial style learning can boost results even surpassing existing state-of-the-art solutions, especially considering the ease of implementation of this idea. The experiments section is quite strong, leveraging well-known datasets, relatively modern segmentation architectures, and very recent baselines.

---

> ### Author Response · Authors · 2021-11-13
> **Response to Reviewer 38TQ**
>
> >We thank the reviewer for the valuable feedback and suggestions for the improvement. We are happy that the reviewer likes our paper and finds this paper achieves promising results. We are also glad that the reviewer appreciates the simplicity and effectiveness of our method. We address the concerns below.
>
> >**Q1:** The t-SNE visualization (Figure 5) doesn’t seem to convey much information regarding what is happening during the training process. I understand it is typical to visualize features spaces in DA/DG papers, but in this case, it is not clear to me that the figure helps make any point. On the contrary, I would argue it distracts from the main point. Figure 5, as mentioned above, doesn’t seem to help with the narrative.
> >
> >**A:** Sorry for the confusion. First, we conjecture that you may think the t-SNE is generated by **SEMANTIC** features? If so, we would like to clarify that the t-SNE is obtained by the **STYLE** features. Second, we would like to further explain the intention of Fig.5. The purpose of Fig.5 is to observe the behavior of AdvStyle during training. From the results of Fig.5, we can find that (1) our AdvStyle can continuously generate new style features that are different from the original distribution and that (2) the new style features have the chance to be located at the distributions of other datasets during the training process. On the other hand, we agree with you that such stage-based visualization can not fully convey the complete behaviors that happened during the training process. If you still think it is not very helpful to explain our work, we will follow your suggestion and remove it in the revision. We would like to hear your kind opinion.
>
>
> >**Q2:** Why focus just on driving scenes? It should be relatively straightforward to evaluate this approach in other types of scenes, for instance, indoor scenes, by leveraging NYU dataset + SceneNet or InteriorNet. This will help make an even stronger point.
> >
> >**A:** We fully agree with you that our approach can be applied to other types of scenes. The main reason for applying AdvStyle on driving scenes is that the image styles are quite different among the different driving datasets, e.g., the road color. This inspires us to adopt AdvStyle on the driving scenes where style variation is an important factor. Following your suggestion, we have looked into the datasets of indoor scenes and found that there also exist style variations. We then conduct experiments on domain generalization for indoor scene segmentation. We use NYUv2 [Ref-1] as the source domain and SUN RGB-D [Ref-2] as the target domain. We use the shared 13 classes for training and testing. The inputs are RGB images. We use the same segmentation model and implementation settings as our paper, i.e., Deeplab-V3+ with ResNet-50. Results are shown in the Table below. Our AdvStyle can also improve the performance of the baseline, but the improvement is not very significant. We conjecture the reason is that style variations are similar in these two datasets. Indeed, our AdvStyle is more suitable for synthetic-to-real DG in the urban scene, where the styles are very different between the synthetic data and the real data. We would like to keep the scope of this paper in the driving scene and leave the study in other types of scenes (such as indoor) to the future work.
> >
> >| Method | mIoU |
> >| -------- | -------- |
> >| Baseline    | 44.4  |
> >| **Baseline + AdvStyle**    | **46.2** |
> >
> >**Reference**
> >
> >[Ref-1] Silberman et al. Indoor Segmentation and Support Inference from RGBD Images. In Proc. ECCV 2012.
> >
> >[Ref-2] Song et al. SUN RGB-D: A RGB-D Scene Understanding Benchmark Suite. In Proc. CVPR 2015.
> >
> >[Ref-3] Zhang et al. Physically-based rendering for indoor scene understanding using convolutional neural networks. In Proc CVPR 2017.
>
>
> >**Q3:** Figure 1 is cluttered and hard to understand. I would recommend removing b) and d).
> >
> >**A:** Thanks for your suggestion. We will remove (b) and (d) of Fig.1 in the next version.

---

### Official Review · Reviewer_9eF2 · 2021-10-31

**Correctness:** 3
**Technical Novelty And Significance:** 3
**Empirical Novelty And Significance:** 2
**Recommendation:** 6
**Confidence:** 5

**Main Review:**

Strength:
- The paper is very well-written and easy to follow
- The proposed method is simple yet effective with a good motivation
- The proposed augmentation can be integrated into different methods
- The paper shows good performance on two tasks: semantic segmentation and image classification

Weakness:
- The explanation of why the proposed augmentation performs well to cover unseen data distributions is not clear. For example, in Fig 5 of the t-SNE plot, GTAV is the only training data, but it can generate out-of-distribution samples (e.g., Adv GTAV), which does not seem very intuitive. Is it also because other data augmentations (color jittering and gaussian blur) are used? The authors should clarify this.

Experimental comparisons
- In Table 5, although there are different settings, the authors should try efforts to make fair comparisons, e.g., using the whole set of GTAV for training (setting I and II).
- The setting in the paper only uses one source dataset. It would be interesting to report results of using multiple source datasets (like the ISW paper using both GTAV and Synthia).

Minor issue
- The ERM baseline in Sec 4.5 should have the reference, instead of having it in the appendix.

##########################################################################

Post-rebuttal:
The rebuttal well addresses my main concerns. I would encourage the authors to release the code and models if the paper is accepted.
##########################################################################

**Summary Of The Paper:**

The paper introduces a domain generalization method for semantic segmentation, where one synthetic data is given as the source training dataset and the model is tested on unseen real datasets. The authors propose a simple yet effective method by introducing adversarial style augmentations. Specifically, the model decomposes the image into a style feature and a normalized image, and then uses the segmentation loss to update the style feature. With the updated style feature and the normalized image, it can resemble an augmented image, which is then used to optimize the segmentation model along with the original image via the segmentation losses. A few highlights of the proposed method are:
- It appears to be a plug-in module, so that it can be used for different methods. architectures, and tasks
- It shows state-of-the-art performance on the semantic segmentation task

**Summary Of The Review:**

Although the proposed augmentation method is simple. it shows state-of-the-art performance for domain generalizable semantic segmentation. The proposed method can be also adopted in different methods, backbones, and tasks, which is beneficial to the field. However, the authors should address the raised concerns regarding the technical clarity and experimental comparisons.

---

> ### Author Response · Authors · 2021-11-13
> **Response to Reviewer 9eF2 (Part I)**
>
> >We thank the reviewer for the valuable feedback and suggestions for the improvement. We are happy that the reviewer finds this paper achieves promising results and believes that this work is beneficial to the field. We are also glad that the reviewer appreciates the simplicity and effectiveness of our method. We address the concerns below. We would highly appreciate it if the reviewer considers upgrading the score based on our feedback.
>
> >**Q1:** The explanation of why the proposed augmentation performs well to cover unseen data distributions is not clear. For example, in Fig 5 of the t-SNE plot, GTAV is the only training data, but it can generate out-of-distribution samples (e.g., Adv GTAV), which does not seem very intuitive. Is it also because other data augmentations (color jittering and gaussian blur) are used? The authors should clarify this.
> >
> >**A:** Sorry for the confusion. We would like to further explain this.
> >
> >First, although the model is trained with AdvStyle and other two augmentations (color jittering and gaussian blur), the adversarial style features in Fig.5 are directly generated from the original images. That is, at each stage, given the current trained model, we use the original images without augmentations to produce the adversarial style features with the adversarial learning. Therefore, these out-of-distribution styles are generated by our AdvStyle instead of the other two augmentations.
> >
> >Second, recall that, with adversarial learning, we are encouraged to generate adversarial style features that are difficult to the current model. Along with the training, the model will be familiar with the style distribution of the training data and most of the style features in this distribution are no longer difficult for the model. In such a context, adversarial learning will encourage us to generate out-of-distribution styles, which are not seen before and difficult to the current model. Also, the generated images of out-of-distribution styles have the chance to be located at the distributions of other datasets. We will include these clarifications in the revision.
> >
> >We also would like to give a toy example to explain this. Suppose the style feature is 1-dimensional, and the style distributions of G (source) and C (target) are in the ranges of [0, 1] and [1, 2], respectively. Once the model is robust to the style feature in the range of [0, 1], the adversarial learning will produce a gradient that leads the style feature away from the range of [0, 1]. For example, the value of the style feature is 0.8 and the gradient generated by adversarial learning is 0.3. Then, we will obtain an ``adversarial style feature = 1.1’’, which is located at the distribution of C $\in$ [1, 2].
>
>
> >**Q2:** In Table 5, although there are different settings, the authors should try efforts to make fair comparisons, e.g., using the whole set of GTAV for training (setting I and II).
> >
> >**A:** Thanks for your valuable suggestion. During rebuttal, we use the whole set of GTAV for training ISW and ISW+AdvStyle. The results with the ResNet-101 backbone are shown below. We can find that using the whole set of GTAV can produce higher results on CityScapes and Maphillary. We are sorry that we could not finish the results of all the models during rebuttal due to limited computation resources. Nevertheless, we promise to update Table 5 in the final version of this work.
> >
> >| Method       | GTAV Set | CityScapes | BDD   | Mapillary | Mean  |
> >|--------------|----------|------------|-------|-----------|-------|
> >| ISW          | Whole    |    37.51   | 33.54 |   36.12   | 35.72 |
> >| **ISW+AdvStyle**| Whole    |    **44.51**   | **39.27** |   **43.48**   | **42.42** |
> >| ISW          | Train    |    37.20   | 33.36 |   35.57   | 35.37 |
> >| **ISW+AdvStyle**| Train   |    **43.44**   | **40.32** |   **41.96**  | **41.90** |

---

> ### Author Response · Authors · 2021-11-13
> **Response to Reviewer 9eF2 (Part II)**
>
> >**Q3:** The setting in the paper only uses one source dataset. It would be interesting to report results of using multiple source datasets (like the ISW paper using both GTAV and Synthia).
> >
> >**A:** Thanks for your suggestion. During rebuttal, we applied our AdvStyle to the setting of multi-source by using both GTAV and Synthia as the source data. Results are shown in the Table below. Clearly, our AdvStyle consistently improves the results of ISW, further verifying the effectiveness of the proposed AdvStyle. Again, due to the limited computation resources, we only report the results of ISW+AdvStyle here. However, we will add the results of other models (baseline and IBN-Net) in our final version.
> >
> >| Method       | Training Set | CityScapes | BDD   | Mapillary | Mean  |
> >|--------------|--------------|------------|-------|-----------|-------|
> >| Baseline     | G+S          | 35.46      | 25.09 | 31.94     | 30.83 |
> >| IBN-Net          | G+S          | 35.55      | 32.18 | 38.09     | 35.27 |
> >| ISW          | G+S          |    37.69   | 34.09 |   38.49   | 36.75 |
> >| **ISW+AdvStyle** | G+S          |     **39.29** | **39.26** | **41.14** | **39.90**|
>
>
> >**Q4:** The ERM baseline in Sec 4.5 should have the reference, instead of having it in the appendix.
> >
> >**A:** Thanks. In our manuscript, we have added the reference of ERM in Sec 4.5.

---

> ### Author Response · Authors · 2021-11-20
> **Thank Reviewer 9eF2 for the review and approval**
>
> We sincerely thank you for your valuable review and reply. We are very glad that you appreciate our response and keep the acceptance score. We will release the code and model as soon as our paper is accepted.

---

### Official Review · Reviewer_8yWf · 2021-11-02

**Correctness:** 2
**Technical Novelty And Significance:** 2
**Empirical Novelty And Significance:** 2
**Recommendation:** 5
**Confidence:** 3

**Main Review:**

Strengths:
1. Complete and solid experiments
2. The image-level style transfer part is intuitive
3. Paper is easy to follow

Weakness:
The main concern I have is the model design and also not sure about some of the authors' intuitions behind:
1. Based on the algorithm in Appendix B, we only will update the adversarial learning once, or the mean and variance once, then we feed the updated mean and variance to robust model part to reconstruct the scene and then further update the segmentation model. My main concern is about what is really learnt in this one update? Personally, I feel that when updating only once, the generator cannot really learn much, e.g. the reconstructed results can be very far away from the original image and cannot be very meaningful. Then if at later stage we force the predictor to minimize the segmentation loss of both original and reconstructed images, will this introduce noise to our predictor, e.g. the predictor then cannot really do well on either image. And is this scenario possible and how does the proposed method handle this problem?
2. Is the segmentation model shared between adversarial and robust model learning process? I believe so, right? Then follow up for the first question, will the segmentation model be drifted away by the one-time update and then introduce additional errors to adversarial learning procedure?
3. How do we decide the maximum number of iterations in algorithm?
4. One corner/extreme case might be that the generator learns very well after many times of iterations and can perfectly reconstruct original images. Then under this scenario, the segmentation model do not need to learn anything but stay with what it has been initialized with would be the right answer for the model. Or we can relax a bit by thinking that what has been generated/reconstructed can be very close to the original images after many epochs and not much changes are needed for segmentation model. How will the proposed method handle this case?
5. What is the difference/relationship between the proposed method and papers like "LEARNING TO SIMULATE"?
6. Why the proposed method can produces new styles that cover the distributions of different datasets as shown in Fig.5? Any explanation?

**Summary Of The Paper:**

This paper aims to address the domain generalization problem in semantic segmentation. Starting to synthetic source dataset, the authors propose to apply image-level style transfer to bridge the gap between source and unseen real dataset. Specifically, the image-level style is defined as the mean and variance of each image. And the proposed method learns to reconstruct scenes with their styles and further updates the segmentation model with both original images and reconstructed images.

In experiments, the authors show that the proposed method can:
1. achieve SOTA performance on multiple real datasets
2. generalize well with various backbones
3. cover the data distributions of real dataset well

**Summary Of The Review:**

In general, I feel that the paper provides very good experimental results but I have concerns about their model design. I am willing to change my rating if the authors can address my concerns above.

---

> ### Author Response · Authors · 2021-11-13
> **Response to Reviewer 8yWf (Part I)**
>
> >We thank the reviewer for the valuable feedback and suggestions for the improvement. We are happy that the reviewer finds this paper achieves promising results. We address the concerns below. We hope that our feedback will increase the reviewer's confidence and we would highly appreciate it if the reviewer considers upgrading the score based on our feedback.
>
>
> >**Q1:** Based on the algorithm in Appendix B, we only will update the adversarial learning once, or the mean and variance once, then we feed the updated mean and variance to robust model part to reconstruct the scene and then further update the segmentation model. My main concern is about what is really learnt in this one update? Personally, I feel that when updating only once, the generator cannot really learn much, e.g. the reconstructed results can be very far away from the original image and cannot be very meaningful. Then if at later stage we force the predictor to minimize the segmentation loss of both original and reconstructed images, will this introduce noise to our predictor, e.g. the predictor then cannot really do well on either image. And is this scenario possible and how does the proposed method handle this problem?
> >
> >**A:** Good question. We would like to further explain our method.
> >* **What is really learnt in AdvStyle in one update?** Given one image, we first update the style features, i.e., mean and variance features, by adversarial learning on the segmentation model. The segmentation model is freezed here and only the style features are updated. Then, we use the learned adversarial style features and the normalized image to generate the adversarial example, which is utilized to train the robust segmentation model together with the original image (the segmentation model is only optimized at this stage). Note that, in the reconstruction stage, the reconstructed image is not generated by the robust model (or called generator) but simply by the de-normalizing operation on the learned adversarial style feature and the normed image. In a nutshell, we learn adversarial style features by adversarial learning in the first stage and learn the robust model with the adversarial and original images in the second stage.
> >
> >* **Will the reconstructed results be very far away from the original image?** Recall that the style features are not sampled from some distributions or generated by a model (like VAE or GAN). Instead, they are initialized by the original style features and updated by adversarial learning, which is actually a small-perturbed version of the original style features. Thus, given a proper learning rate (we experimentally set it to 3 in this paper), the style feature will not be very far away from the original style feature and thus the reconstructed image will be meaningful in most cases. Following your concern, during rebuttal, we visualized the reconstructed images through training and found that the model seldom produces aberrant (unmeaningful) style features. The probability of such bad cases is very low and they only occur in the early training stages (at first 3 epochs). To handle this problem, we set a threshold to discard such outliers during training. Specifically, when the L1-distance between the generated style feature and the mean style feature of the training data is larger than 3, we discard such adversarial style features during training. We find that we achieve very similar results to our original implementation. Therefore, we believe these outliers have very limited influence on the segmentation model and we can ignore them during training.
>
> >**Q2:** Is the segmentation model shared between adversarial and robust model learning process? I believe so, right? Then follow up for the first question, will the segmentation model be drifted away by the one-time update and then introduce additional errors to adversarial learning procedure?
> >
> >**A:** Right, the segmentation model is shared between two processes. However, as explained in our reply of Q1, for the segmentation model, it is freezed when updating style features by adversarial learning (the ``one-time update’’), and is only optimized by the original and the reconstructed images in the robust model learning stage. For the reconstructed images, the outliers have very limited influence on the segmentation model. Based on the two reasons, it is stable in training the segmentation model with our AdvStyle.
>
> >**Q3:** How do we decide the maximum number of iterations in algorithm?
> >
> >**A:** For fair comparison, we follow ISW [Choi et al., 2021] to set the number of iterations to 40K, which has been empirically demonstrated to achieve suitable results for all models.

---

> ### Author Response · Authors · 2021-11-13
> **Response to Reviewer 8yWf (Part II)**
>
> >**Q4:** One corner/extreme case might be that the generator learns very well after many times of iterations and can perfectly reconstruct original images. Then under this scenario, the segmentation model do not need to learn anything but stay with what it has been initialized with would be the right answer for the model. Or we can relax a bit by thinking that what has been generated/reconstructed can be very close to the original images after many epochs and not much changes are needed for segmentation model. How will the proposed method handle this case?
> >
> >**A:** We would like to clarify two matters. First, as explained in our reply of Q1, in the reconstruction stage, the reconstructed image is NOT generated by the robust model (or called generator) but simply by the de-normalizing operation on the learned adversarial style feature and the normed image. Second, in our AdvStyle, we are NOT trying to reconstruct the original image but to generate an adversarial image that is difficult to be recognized by the segmentation model. The generated adversarial image keeps the semantic content of the original image but in a different style, which thus will NOT be the same as the original image. Moreover, the adversarial styles are generated online at each iteration, based on the increase of segmentation loss. Therefore, the adversarial images will always be novel even after many epochs, which can be continuously used to improve the generalization ability of the model. Taking the above explanations, our AdvStyle does not meet the case mentioned by the Reviewer 8yWf.
>
> >**Q5:** What is the difference/relationship between the proposed method and papers like "LEARNING TO SIMULATE"?
> >
> >**A:** This work and [Ref-1] are different in two aspects.
> >* **Different Tasks**: [Ref-1] aims to learn good sets of parameters for an *image rendering simulator* in actual computer vision applications. Instead, this work aims to learn a *generalized model* for semantic segmentation.
> >* **Different Optimizations**: [Ref-1] controls the distribution of synthesized data to *maximize the accuracy* of a model trained on that data. In contrast, this work uses adversarial learning to generate stylized images that *are difficult to* the segmentation model.
> >
> >**Reference**
> >
> >[Ref-1] Learning To Simulate. Nataniel Ruiz, Samuel Schulter, Manmohan Chandraker. In Proc ICLR 2019.
>
>
> >**Q6:** Why the proposed method produce new styles that cover the distributions of different datasets as shown in Fig.5? Any explanation?
> >
> >**A:** Sorry for the confusion. We would like to further explain this. Recall that, with adversarial learning, we are encouraged to generate adversarial style features that are difficult to the current model. Along with the training, the model will be familiar with the style distribution of the training data and most of the style features in this distribution are no longer difficult for the model. In such a context, adversarial learning will encourage us to generate out-of-distribution styles, which are not seen before and difficult to the current model. Also, the out-of-distribution styles have the chance to be located at the distributions of other datasets. We will include these clarifications in the revision.
> >
> >We also would like to give a toy example to explain this. Suppose the style feature is 1-dimensional, and the style distributions of G (source) and C (target) are in the ranges of [0, 1] and [1, 2], respectively. Once the model is robust to the style feature in the range of [0, 1], the adversarial learning will produce a gradient that leads the style feature away from the range of [0, 1]. For example, the value of the style feature is 0.8 and the gradient generated by adversarial learning is 0.3. Then, we will obtain an ``adversarial style feature = 1.1’’, which is located at the distribution of C $\in$ [1, 2].

---

### Official Review · Reviewer_FUQj · 2021-11-02

**Correctness:** 3
**Technical Novelty And Significance:** 2
**Empirical Novelty And Significance:** 2
**Recommendation:** 5
**Confidence:** 5

**Main Review:**

Strengths:
* This paper tackles a structured prediction problem (semantic segmentation) in the domain generation community.
* The idea is clearly stated and straightforward with source code included in the appendix.

Weaknesses:
* The technical novelty is very limited. The key idea (augmenting channel-wise mean and standard deviation) has been explored in the cited reference [MixStyle; Zhou et al., 2021]. The difference is that this paper constrains the augmentation within the first layer with an adversarial formulation. The reviewer fails to perceive a broader impact on the community from this technical contribution.
* At a high level, AdvStyle is a data augmentation method that generates unconstrained adversarial examples. In this sense, a prior work [**Ref1, Bhattad, et al., In ICLR 2020] is highly-related but not discussed in the paper. Bhattad et al. use a pre-trained colorization or texture-transfer network for adversarial example generation. Please elaborate on this in the rebuttal and incorporate the comparisons in the future version of the paper.
* This paper only conducts experiments on sim-to-real transfer but does not evaluate real-to-sim transfer or real-to-real transfer. This makes an unfair comparison to the baseline methods including [Choi et al., 2021]. Please elaborate on this in the rebuttal. The reviewer would kindly suggest incorporating these comparisons in the future version of the paper.

References:

* Ref1: Unrestricted adversarial examples via semantic manipulation. Bhattad et al., In ICLR 2020.

**Summary Of The Paper:**

This paper studies the problem of adversarial domain randomization for urban scene semantic segmentation. Specifically, it proposes an adversarial augmentation method called AdvStyle that injects adversarial signals into the channel-wise mean and standard deviation of images. The style augmentation is used when training on synthetic images from Synthia and GTAV datasets. This paper demonstrates promising results when applied the trained model to real semantic segmentation benchmarks.

**Summary Of The Review:**

Overall, this is a borderline paper with promising results. The major concerns include the limited technical novelty, missing references, and unfair experimental comparisons. I would like to see the rebuttal and comments from the other reviewers to make a final decision.

---

> ### Author Response · Authors · 2021-11-13
> **Response to Reviewer FUQj (Part I)**
>
> >We thank the reviewer for the valuable feedback and suggestions for the improvement. We are happy that the reviewer finds this paper achieves promising results. We address the concerns below. We would highly appreciate it if the reviewer considers upgrading the score based on our feedback.
>
> >**Q1:** The technical novelty is very limited. The key idea (augmenting channel-wise mean and standard deviation) has been explored in the cited reference [MixStyle; Zhou et al., 2021]. The difference is that this paper constrains the augmentation within the first layer with an adversarial formulation. The reviewer fails to perceive a broader impact on the community from this technical contribution.
> >
> >**A:** We would like to further emphasize our contributions.
> >* **New Style-based Augmentation Method**: Although this work and MixStyle both use the style feature (channel-wise mean and standard deviation) to achieve data augmentation, they are very different in implementation (acknowledged by Reviewer FUQj). Specifically, MixStyle mixes the styles of two samples with a convex weight while this work augments the styles in an adversarial learning manner. We think this new augmentation manner, which is very different from MixStyle, should be considered as a novelty of this work.
> >* **Promising Results**: As acknowledged by all the reviewers, our AdvStyle achieves large improvements on different backbones, methods and tasks. This is beneficial to the field (acknowledged by Reviewer 9eF2 and Reviewer 38TQ). In addition, we show that our AdvStyle produces clearly higher results than the MixStyle (in Table 4), demonstrating the advantage of the proposed augmentation manner.
> >* **Easy to Implement**: Our AdvStyle is easy to implement (acknowledged by Reviewer 9eF2 and Reviewer 38TQ), which can be readily applied on different backbones, methods and tasks without a significant modification.
> >
> >Based on the above explanations, we hope the reviewer FUQj could find that our paper proposes a **new, effective and easy-to-use** approach, making a non-trivial step toward the problem of domain generalization in semantic segmentation.
>
>
> >**Q2:** At a high level, AdvStyle is a data augmentation method that generates unconstrained adversarial examples. In this sense, a prior work [**Ref1, Bhattad, et al., In ICLR 2020] is highly-related but not discussed in the paper. Bhattad et al. use a pre-trained colorization or texture-transfer network for adversarial example generation. Please elaborate on this in the rebuttal and incorporate the comparisons in the future version of the paper.
> >
> >**A:** We fully agree with you that AdvStyle can be regarded as a data augmentation method. We also thank you for pointing out this related work [Ref1]. Below we compare the differences between AdvStyle and [Ref1], which are reflected in three aspects:
> >* **Different Tasks**: [Ref1] aims to learn adversarial examples that can largely impair the performance of models. In contrast, AdvStyle aims to produce adversarial examples that can be used to improve the generalization ability of the segmentation model.
> >* **Different Coloring/Stylizing Methods**: [Ref1] uses a pre-trained colorization network to color the images. Instead, AdvStyle exploits the channel-wise mean and standard deviation to stylize the images.
> >* **Different Implementation Complexities**: To obtain the adversarial examples, [Ref1] requires a pre-trained colorization network, and introduces several hyper-parameters, such as learning rate, number of input hints, and number of clusters. Instead, we directly optimize the 6-dim style feature with a fixed learning rate, which is more easy to implement.
> >
> >**Implementing [Ref1] in our task**: During rebuttal, we injected [Ref1] into our framework by replacing our AdvStyle with [Ref1]. We use the default parameter settings provided by [Ref1]. The number of adversarial iteration is set to 1 as our AdvStyle. However, we find that [Ref1]-based framework is much slower than our AdvStyle-based framework. Specifically, [Ref1]-based framework requires 15 times more training time than our AdvStyle-based framework. In addition, the training GPU memory is significantly increased for the [Ref1]-based framework, which is 2 times more than ours. Therefore, [Ref1] is not suitable for training domain generalized segmentation models, considering the efficiency. This further demonstrates the advantage of our AdvStyle.
> >>
> >Taking the above explanations, we hope the reviewer could find the differences between AdvStyle and [Ref1], especially in the aspects of task and ease of use. In addition, we hope the reviewer could find the advantage of our AdvStyle over [Ref1] in addressing the problem of domain generalized semantic segmentation. We will include the above comparisons in the revision.
> >
> >**Reference**
> >[Ref1] Unrestricted adversarial examples via semantic manipulation. Bhattad et al., In ICLR 2020.

---

> ### Author Response · Authors · 2021-11-13
> **Response to Reviewer FUQj (Part II)**
>
> >**Q3:** This paper only conducts experiments on sim-to-real transfer but does not evaluate real-to-sim transfer or real-to-real transfer. This makes an unfair comparison to the baseline methods including [Choi et al., 2021]. Please elaborate on this in the rebuttal. The reviewer would kindly suggest incorporating these comparisons in the future version of the paper.
> >
> >**A:** *First*, we should emphasize that we DO NOT make an unfair comparison to the baseline methods. In the sim-to-real scene, we compare our method with the baselines under exactly the same settings, i.e., the backbone, training data, data augmentations, learning rate, batch size and optimizer are the same.
> >
> >*Second*, we would like to reiterate that the main goal of this paper is ``synthetic-to-real DG for semantic segmentation’’. The main reasons for studying the synthetic-to-real DG are that (1) the synthetic data can be automatically generated and annotated, which can significantly lower the cost, and that (2) the use of the synthetic data can largely protect data privacy which is recently repeatedly underlined in the community.
> >
> >Taking the above explanations, we hope the reviewer could find that we make a fair comparison to the baselines and find the motivation and advantage of studying our method in the synthetic-to-real DG scene.
> >
> >On the other hand, to make the comparison of this paper more consistent with [Choi et al., 2021] under the synthetic-to-real scene, during rebuttal, we conducted experiments on the multi-source setting, where GTAV and Synthia were used as the training data. Results are shown in the Table below. Clearly, our AdvStyle consistently improves the results of ISW, further verifying the effectiveness of the proposed AdvStyle. We are sorry that we could not finish the results of all the models during rebuttal due to limited computation resources. However, we promise to add the results of other models (baseline and IBN-Net) in our final version.
> >
> >| Method       | Training Set | CityScapes | BDD   | Mapillary | Mean  |
> >|--------------|--------------|------------|-------|-----------|-------|
> >| Baseline     | G+S          | 35.46      | 25.09 | 31.94     | 30.83 |
> >| IBN-Net          | G+S          | 35.55      | 32.18 | 38.09     | 35.27 |
> >| ISW          | G+S          |    37.69   | 34.09 |   38.49   | 36.75 |
> >| **ISW+AdvStyle** | G+S          |     **39.29** | **39.26** | **41.14** | **39.90**|

---

### Author Response · Authors · 2021-11-22
**Summary of Revision**

Dear Reviewers,

We sincerely thank you for the valuable feedback and suggestions for improvement. According to your suggestions, we have carefully revised our manuscript. The modifications are summarized below.

* Redraw Fig.1 (in Section 1, Page 2). Suggestion from Reviewer 38TQ.
* Provide a discussion on two related works ([Ref-1] and [Ref-2], in Section 2, Page 3). Suggestion from Reviewer FUQj and Reviewer 8yWf.
* Add new experiments, including (1) training with the whole GTAV (in Section 4.3, Page 7-8) and (2) multi-source setting (in Section C, Page 13-14). Suggestion from Reviewer FUQj and Reviewer 9eF2.
* Provide a more detailed explanation for Fig.5 (Section 4.4, Page 8). Suggestion from Reviewer 8yWf, Reviewer 9eF2 and Reviewer 38TQ.
* Include the reference for ERM (in Section 4.5, Page 9). Suggestion from Reviewer 9eF2.

**Reference**

[Ref-1] Unrestricted adversarial examples via semantic manipulation. Bhattad et al., In ICLR 2020.

[Ref-2] Learning To Simulate. Nataniel Ruiz, Samuel Schulter, Manmohan Chandraker. In Proc ICLR 2019.

---

### Decision · Program_Chairs · 2022-01-20

**Decision:**

Reject

**Comment:**

This paper presents a domain generalization method for semantic segmentation. The model is trained on synthetic data (source) and is tested on unseen real datasets (target). The authors propose a simple data augmentation method, AdvStyle, generating unconstrained adversarial examples for the training on the source domain.

There was no consensus on the method among the reviewers. Several issues have been raised. After rebuttal and discussion, no one really changed her/his mind. The motivation of why focus just on driving scenes is still questionable. Definitively, it could be interesting to investigate further why it is not straightforward to have gains on other kinds of scenes. Finally, we encourage the authors to address the raised concerns regarding the discussion with previous works and the comparisons for future publication.